# Understanding Molecular Mechanisms of Phenotype Switching and Crosstalk with TME to Reveal New Vulnerabilities of Melanoma

**DOI:** 10.3390/cells11071157

**Published:** 2022-03-29

**Authors:** Ahmad Najem, Laura Soumoy, Malak Sabbah, Mohammad Krayem, Ahmad Awada, Fabrice Journe, Ghanem E. Ghanem

**Affiliations:** 1Laboratory of Clinical and Experimental Oncology (LOCE), Institut Jules Bordet, Université Libre de Bruxelles, 1000 Brussels, Belgium; ahmad.najem@bordet.be (A.N.); laura.soumoy@umons.ac.be (L.S.); malak.sabbah@ulb.be (M.S.); mohammad.krayem@bordet.be (M.K.); ahmad.awada@bordet.be (A.A.); fabrice.journe@bordet.be (F.J.); 2Department of Human Anatomy and Experimental Oncology, Université de Mons, 7000 Mons, Belgium; 3Department of Medical Oncology, Institut Jules Bordet, Université Libre de Bruxelles, 1000 Brussels, Belgium

**Keywords:** melanoma, phenotype switching, tumor microenvironment, oxidative stress, metabolic reprogramming, therapeutic strategies

## Abstract

Melanoma cells are notorious for their high plasticity and ability to switch back and forth between various melanoma cell states, enabling the adaptation to sub-optimal conditions and therapeutics. This phenotypic plasticity, which has gained more attention in cancer research, is proposed as a new paradigm for melanoma progression. In this review, we provide a detailed and deep comprehensive recapitulation of the complex spectrum of phenotype switching in melanoma, the key regulator factors, the various and new melanoma states, and corresponding signatures. We also present an extensive description of the role of epigenetic modifications (chromatin remodeling, methylation, and activities of long non-coding RNAs/miRNAs) and metabolic rewiring in the dynamic switch. Furthermore, we elucidate the main role of the crosstalk between the tumor microenvironment (TME) and oxidative stress in the regulation of the phenotype switching. Finally, we discuss in detail several rational therapeutic approaches, such as exploiting phenotype-specific and metabolic vulnerabilities and targeting components and signals of the TME, to improve the response of melanoma patients to treatments.

## 1. Introduction

Metastatic melanoma is notoriously one of the most difficult cancers to treat. The MAPK pathway is constitutively activated in the vast majority (90%) of cutaneous melanoma. BRAF (50%) and NRAS (20–30%) are the most frequent mutations followed by NF1 (10–14%) and KIT (5–10%). These genomic alterations are considered driver mutations in melanoma development [1,2]. MAPK inhibitors and immune checkpoint inhibitors have revolutionized the treatment of metastatic melanoma. Despite these recent advances, many patients do not respond to these therapies, and the acquired resistance remains a major problem [1,2].

The acquired resistance can be driven by genetic alterations activating the MAPK pathway, such as BRAF amplification, expression of BRAF splicing variants, NRAS and MEK mutations, or by MAPK-independent alterations, such as the activation of the PI3K/AKT pathway (PIK3CA mutations, AKT mutations/amplifications, and PTEN loss), and cyclin D1 amplifications [1,2]. The intertumoral and intratumoral genetic heterogeneities are also major obstacles to targeted therapy. Indeed, the emergence of distinct subclones with various mutations, including drug-resistant ones within the tumor or among different tumor sites, can lead to disease relapse [3].

However, melanoma cells are not only able to adapt to therapies by acquiring genetic alterations but they can also switch their cellular phenotype in order to adapt to therapeutics and various stressful conditions. This phenotypic plasticity, which has gained more attention in cancer research, is proposed as a new paradigm for melanoma progression and resistance to therapy. Such plasticity, referred to as phenotype switching in melanoma, is linked to epithelial–mesenchymal transition (EMT) and is characterized as dynamic cell state transitions that involve reversible transcriptional changes and epigenetic modifications in contrast to the irreversible genetic alterations [4,5,6,7]. Phenotypic plasticity is an important source of tumor heterogeneity and presents a major challenge for both targeted and immunotherapies.

The phenotype switching model in melanoma was initially described by Hoek et al. (2006) based on gene expression profiling data from both cell lines and melanoma patient samples [5,7]. This model is controlled by MITF a master lineage transcription factor and EMT transcription factors (TFs). According to this model, melanoma cells can interconvert between two main phenotypes, the melanocytic (differentiated/proliferative) state and the mesenchymal-like (invasive/undifferentiated) state [5,7].

In this review, we will discuss in detail: (1) the complex spectrum of phenotype switching and the variety of melanoma states; (2) the epigenetic and metabolic reprogramming involved in phenotypic plasticity; and (3) the prominent role of the tumor microenvironment and oxidative stress in the regulation of phenotype switching. Moreover, we will discuss therapeutic opportunities based on melanoma states vulnerabilities, metabolic rewiring, and microenvironment modulation.

## 2. Phenotype Switching in Melanoma: Growing Ever More Complex

### 2.1. Key Regulator Factors of Phenotype Switching

Melanoma cells can interconvert between two main states: the proliferative/differentiated MITF^high^ phenotype, also referred to as the “melanocyte-like” state, and the invasive/undifferentiated MITF-low phenotype, referred to as the “mesenchymal-like” state.

#### 2.1.1. The MITF “Rheostat Model”

MITF (Microphthalmia-associated transcription factor) is the master regulator transcription factor in melanoma involved in the regulation of melanogenesis, as well as cell proliferation and survival by regulating several genes implicated in differentiation, survival, and metabolism [8,9,10]. Indeed, MITF functions as a rheostat that dictates the phenotype of melanoma cells. Low and very low levels of MITF are associated, respectively, with senescence and invasion, while high and very high levels are associated, respectively, with proliferation and differentiation [8,9,10]. 

#### 2.1.2. ZEB2 and ZEB1 in Melanoma Plasticity

MITF interconnects with EMT to control the phenotypic plasticity in melanoma cells. EMT-TF factors involve several protein families, including the ZEB family [11,12] that comprises two proteins: zinc finger E-box binding protein 1 (ZEB1) and zinc finger E-box binding protein 2 (ZEB2) [11,12]. Denecker et al. [12] identified that the ZEB2–MITF–ZEB1 transcriptional network controls melanogenesis and melanoma progression. They demonstrated that ZEB2 regulates MITF and thereby controls differentiation of the melanocyte lineage. Indeed, the loss of ZEB2 in the melanocyte lineage resulted in a downregulation of MITF and melanocyte differentiation markers concomitant with an upregulation of ZEB1 [12]. In accordance with these data, it was shown that phenotype switching towards an invasive state is associated with ZEB1 activation [13]. 

#### 2.1.3. Opposing Roles of MITF and Wnt5A in Phenotype Switching

The reciprocal antagonism between MITF and Wnt5A can also control the dynamic phenotype switching in melanoma cells [14]. Indeed, an increased expression of Wnt5A is associated with a decreased level of melanocytic markers and a dedifferentiated state [15]. 

#### 2.1.4. Role of MITF–BRN2 Axis in Phenotype Switching

BRN2, also referred to as class III POU domain protein (POU3F2), is a neural-lineage transcription factor involved in melanocytic development [16,17]. The MITF–BRN2 axis also has a central role in the regulation of phenotype switching. Indeed, MITF and BRN2 expression are inversely correlated and mark two distinct subpopulations with two opposing phenotypes. While MITF^high^ cells display a differentiated phenotype, high BRN2 expression marks the mesenchymal-like phenotype [16,17]. BRN2 acts as a counterbalance factor to MITF and vice versa. Indeed, BRN2 leads to the repression of MITF transcription, while MITF reduces BRN2 levels via the expression of miR-211 [18].

BRN2 is considered a master modulator of the invasive/undifferentiated phenotype. Indeed, overexpression of BRN2 leads to a less differentiated phenotype in melanocytes and promotes a phenotypic switch towards an invasive phenotype in melanoma cells [16]. Furthermore, several drivers of the invasion were identified as potential direct downstream targets of BRN2, such as ABCB1, ABCB5, CD36, and CDH-13 [16]. Among BRN2 target genes, it was shown that NFIB can mediate BRN2-driven melanoma invasion through dynamic changes in the chromatin state of melanoma cells [19].

#### 2.1.5. Antagonistic Functions of SOX10 and SOX9 in the Regulation of Phenotype Switching

SOX9 and SOX10 show antagonistic functions in the regulation of melanoma phenotype [4,20]. These genes belong to the SOX (Sry (sex determining region Y)-related HMG box) family that plays a critical role in embryonic development and the regulation of cell fate [20]. The melanocyte-like state (proliferative/differentiated phenotype) shows high expression of SOX10 and the mesenchymal-like state (invasive/undifferentiated) shows high expression of SOX9. Indeed, SOX10 positive melanoma cells show an upregulation of genes implicated in pigmentation and cell differentiation while SOX9 positive melanoma cells show an upregulation of genes implicated in EMT [4]. Therefore, SOX10 is a main regulator of the melanocytic state while SOX9 is a main regulator of the mesenchymal-like phenotype.

#### 2.1.6. RTKs as Putative Drivers of Phenotype Switching

The phenotype switching in melanoma is marked with downregulation of MITF and upregulation of RTKs, such as AXL and EGFR [5,21,22]. RTKs are major drivers of the phenotype switching towards the mesenchymal-like state and high RTK expression marks the undifferentiated MITF low population [5,21,22,23]. 

MITF and AXL have been identified irrespective of BRAF and NRAS mutations as part of two opposing gene expression patterns [5,21,22,23]. MITF^high^/AXL^low^ and MITF^low^/AXL^high^ populations contribute mostly to intratumor heterogeneity in melanoma and, thereby, resistance to therapy [5,21,22,23]. Furthermore, the expression of EGFR and MITF are inversely correlated in melanoma and forced expression of MITF in melanoma and colon cancer cells inhibits EGFR expression [24]. EGFR, along with SOX9, are upregulated in melanoma cells with undifferentiated phenotypes, and their expressions are induced by the loss of SOX10 [25].

Moreover, it was demonstrated that the phenotype switching towards an invasive state is associated with upregulation of FGF2 and ROR2 [26,27]. Upregulation of ROR2 is critical for the activation of the Wnt5A pathway [26,27]. On the other hand, FGF2 activates the FGFR1 receptor and can promote cell migration via downregulation of focal adhesion kinase (FAK) [28].

#### 2.1.7. BORIS Mediates TGF-β-Driven Phenotype Switching

TGF-β is a well-known inducer of EMT and invasiveness in cancer and it was identified as a potent repressor of MITF [29,30]. Indeed, TGF-β represses MITF via the inhibition of PKA and it acts as an inducer of GLI2, a negative regulator of MITF [30]. Recently, it was demonstrated that the brother of the regulator of imprinted sites (BORIS) promotes phenotype switching in melanoma cells via activation of TGF-β signaling. BORIS plays a role as an invasion-promoting transcriptional regulator and it was reported to be upregulated in melanoma and implicated in melanoma progression [29].

#### 2.1.8. TEADs as Key Regulators of the Invasive Phenotype

Transcriptional enhanced associate domain (TEAD) transcription factors play a critical role in tumorigenesis and they have emerged as key drivers of cancer progression, EMT, and metastasis [5,31]. The expression of TEADs is associated with poor clinical outcomes and confers invasive and migration potential in several cancers. TEADs are key effectors of the Hippo pathway and they coordinate with other signaling pathways, including Wnt and TGF-β [5,31]. 

Transcriptome analysis in melanoma reveals TEADs as a master regulator of the invasive phenotype [5]. TEAD knockdown inhibits invasion in melanoma cells with an invasive phenotype and downregulates TEAD target genes, including SOX9/Hippo pathway genes, and many other genes linked to metastases promotion [5]. 

TEADs stimulate the upregulation of ZEB1 and drive metastatic squamous carcinoma [32]. Furthermore, TGF-β-induced TEADs promote metastatic phenotypes in breast cancer [33]. Thus, interactions among TEADs, TGF-β, and ZEB1 appear to regulate the invasive phenotype in cancer.

#### 2.1.9. Gene Regulatory Networks 

The two main phenotypes described in melanoma are governed by distinct gene regulatory networks (GRNs). The melanocyte-like phenotype is characterized by the expression of melanocytic lineage transcription factors—MITF, SOX10, ZEB2—and downstream markers, such as TYR, TYRP-1, and melan-A, implicated in cell differentiation and pigmentation, while the mesenchymal-like phenotype is defined by the expression of TEADs, RTK (AXL, EGFR), SOX9, ZEB1, BRN2, and genes involved in the Wnt5A and TGF-β signaling pathways implicated in cell invasion (Figure 1). The tight coordination, the interactions among these factors, and the interplay with the microenvironmental cues control the phenotypic diversity and drive cell state transition.

### 2.2. Refining Classification of Melanoma Phenotypes

Tsoi et al. (2018) showed in their study [25] that, based on gene expression profiling from a panel of human melanoma cell lines, the proliferative and the invasive phenotypes described initially by Hoek [7] can be sub-classified into four melanoma subtypes: the undifferentiated subtype (C1), the neural crest-like subtype (C2), the transitory subtype (C3), and the melanocytic subtype (C4) (Figure 1).

The undifferentiated subtype (C1): is characterized by the expression of invasion and inflammation genes. 

The neural crest-like subtype (C2): shares the invasive/inflammation related signature, but this subtype showed a unique enrichment for neural crest-related gene sets. 

The transitory subtype (C3): a mixed neural-like to melanocyte state. Indeed, this subtype displays a concurrent enrichment of neural crest and pigmentation-associated genes.

The melanocytic subtype (C4): the most differentiated and defined by a strong enrichment for pigmentation-related gene sets.

C1 and C2 both have low MITF levels, but they are distinguished by the SOX10 expression. C1 displays lower levels of SOX10 and a neural crest marker (NGFR) and high expressions of SOX9 and EGFR, indicating an even more undifferentiated state. C3 and C4 are defined by high expressions of MITF and activation of the canonical Wnt/β catenin pathway, supporting a more melanocyte-like signature. Nevertheless, C4 is characterized by a stronger enrichment of MITF target genes implicated in cell pigmentation, indicating a higher degree of differentiation.

C1 and C2 subtypes are related to Hoek’s invasive cohort (referred to as cohort C) while the subtype C4 is related to Hoek’s proliferative one (cohort A). Noteworthy, the transitory subtype (C3) is linked to a cohort termed “B”, identified by Hoek, and shares the characteristics of both proliferative and invasive phenotypes (Figure 1). 

The identification of these semi-invasive melanoma cells (transitory phenotype) that combine the properties of both states with the features of neural crest-like cells, raises the question as to whether this phenotype is evolving as a result of mixed subpopulations reflecting the two different phenotypes in a well-defined manner, or due to switching between these states.

A recent study by Jasper et al. (2020) [4] identified that intermediate transcriptomes are likely due to a stable mixed gene regulatory network. This mixed GRN is specific for the intermediate state.

### 2.3. New Intermediate Cell State and Major Regulators 

Exhaustive analysis of melanoma cell state diversity indicates that, between the two extreme melanocytic and mesenchymal states, there exists an intermediate state. This transition state displays an intermediate migration potential. 

The intermediate state shares several regulators with the melanocyte-like phenotype (MITF, SOX10, and IRF4) and the mesenchymal-like phenotype (AP1, IRF/STAT). However, this state is marked by the activation of specific transcription factors, such as the early growth response 3 (EGR3), NFATc2, and RXRG. Of note, EGR3 is reported as the master regulator of this intermediate cell state and controls the expression of the two other transcription factors.

EGR3 is involved in neurogenesis (immediate and early events), [34] in inflammation, including cytokines release (IL8 and IL6) [35], as well as in angiogenesis, where it has a critical role [36]. EGR3 is also associated with invasion and metastases in various cancers, including prostate [35], breast [37], and lung [38]. Altogether, these data suggest an important role of EGR3 in early transition to an invasive phenotype.

Moreover, calcineurin/nuclear factor of activated T cells C2 (NFATC2) is implicated in tumor angiogenesis growth and invasion [39,40,41]. NFATC2 promotes tumor-initiating phenotypes in lung cancer and represses differentiation [39]. Furthermore, NFATC2-mediated activation of Ets1 is involved in breast cancer invasiveness and EMT [41]. In melanoma, NFATC2 promotes dedifferentiation, immune escape, and cell survival [40,42]. Thus, NFATC2 activation can maintain cell survival under stress conditions leading to the initiation of the undifferentiated/invasive phenotype.

Finally, the nuclear receptor retinoid X receptor gamma (RXRG) is a key driver of the neural crest stem cell (NCSC) transcriptional state involved in the minimal residual disease of melanoma [43] (described below).

Overall, “the intermediate phenotype” is characterized by the expression of key transcription factors implicated in the transition towards a more aggressive phenotype and the acquisition of metastatic potential, indicating a crucial role for this GRN in the progression toward metastatic disease possibly caused by different oxidative stress conditions. 

Importantly, the very existence of the intermediate cell state was validated by in silico analyses of public databases of melanoma biopsies.

### 2.4. Melanoma Cell States Associated with Minimal Residual Disease (MRD)

Many patients achieve astonishing responses to targeted therapies, but after various periods, the majority of them show resistance and relapse. The latter is driven by a small population of residual cells, referred to as “minimal residual disease” (MRD). In their study, Rambow et al. 2018 [43] identified, through single-cell profiling methods, four drug-tolerant transcriptional states that co-occur within MRD (Figure 2): the pigmented state, the starved-like melanoma cell state (SMC), the neural crest stem cell (NCSC), and the invasive state.

The authors showed that drug exposure promotes a transient switch from proliferative to SMC cells that is distinct from other MITF intermediate cells displaying the classical “proliferative” gene signature. SMC shares features of nutrient-deprived cells as well as those associated with both proliferation and invasion. SMC is mainly associated with the high expression of PAX3 and CD36.

Paired box 3 (PAX3) is a key developmental regulator of melanocyte progenitors and CD36 is essential for fatty acid uptake, facilitating proliferation and metastases. PAX3 expression contributes to cell survival and growth in melanoma [44] by maintaining MITF levels in an intermediate range [45] and CD36 through its main role in energy metabolism. Of note, CD36 is a useful marker of melanoma cell adaptation, e.g., to MAPK inhibitors [46].

Following the appearance of the SMC state, cells then undergo a transition towards either a pigmented/differentiated phenotype or towards two undifferentiated phenotypes: the classical invasive state or the NCSC state (Figure 2).

The pigmented state is characterized by high expression of MITF and its downstream target genes involved in cell pigmentation.

The invasive state and the NCSC state have both low levels of MITF but can be distinguished by quite different SOX10 expressions that are higher in NCSC. The NCSC state is also characterized by the expression of neural stem cell markers, such as NGFR and AQP1, and importantly, by an RXR signaling that is identified as the key driver of this state. The invasive and NCSC states appear to be similar to C1 and C2 subtypes described by Tsoi et al. (2018), while the SMC state is related to the C3 transitory subtype.

All of the above point to multiple transcriptional states associated with the MRD, so that melanoma cells can use different trajectories to adapt and survive, further putting forward the prominent role of such phenotypic diversity in melanoma progression and resistance to various therapies.

### 2.5. Senescence-like State

In addition to the proliferative, intermediate, and the invasive states that are defined by high, intermediate, and low levels of MITF, respectively, a “senescence-like state” with extremely high or low levels of MITF has been singled out as an outlier phenotype (Figure 1).

The senescence-like phenotype can be induced in response to various stress signals from TME or upon exposure to cancer therapies and it is proposed as an escape route for tumor cells to evade therapies by allowing prolonged survival in a dormant state with the capacity to recover under microenvironmental signals that promote tumor growth [47].

In this context, Bai et al. 2019 [48] recently proposed a model in which the development of resistance to both targeted therapy and immunotherapy in melanoma includes three phases: (1) early survival (including persister cells and innate resistant cells); (2) reversal of senescence; and (3) new homeostasis.

In the early survival phase, the “senescence-like state” has been discerned among the persisters. This state is associated with the expression of senescence markers and induction of inflammatory network. This phenotype allows the cells to survive under stress conditions and therapies. Senescent cells can expand over time through the generation of a pro-tumorigenic microenvironment. Indeed, they release a variety of cytokines (such as IL6, IL8, and CCL2) and growth factors (such as EGF and FGF) that are termed the senescence-associated secretory phenotype (SASP) [49], promoting TME to shift into a resistance-promoting niche. 

In the second phase (reversal of senescence), the senescence-like state gradually subsides but does not disappear. MITF expression shifts from the extremes back towards a new homeostatic point. MAPK pathway is reactivated and de novo epigenetic and genetic aberrations start to occur during this phase following the chronic treatment with MAPK targeted agents. This phase is highly dynamic, and it is a transitory period that leads to the establishment of new homeostasis (phase 3). 

During the latter, the accumulation of both epigenetic and genetic aberrations and the modulation of the TME result in the establishment of a new homeostasis and irreversible resistance. This phase is associated with high heterogeneity allowing cells persisting under treatment to follow different trajectories.

Therefore, a better understanding of such a senescence-like state and its interplay with the microenvironment is an important key to improve therapeutic strategies.

### 2.6. Hyperdifferentiated State

In a recent study, we identified that the highly differentiated pigmented phenotype observed under treatment with MAPK inhibitors can also occur in drug naïve malignant cells. This state is characterized by the expression of genes implicated in cell differentiation (e.g., TMPRSS13, MAL, MLANA), melanocyte transcriptional programs (e.g., ST8SIA6, HTR2B, MX2, CA12), pigmentation (e.g., TYR, TYRP1, TRPM1, OCA2, S100B, APOD, CCL18, MLC1), and retinal pigment epithelium (e.g., MAMDC2, TFPI2, ASPA, CDH3, SERPINF1) [50].

### 2.7. Deep Characterization of Melanoma Phenotypic Diversity: Lessons to Other Cancers

Altogether, the detailed description of melanoma phenotypes above shows a high degree of phenotypic diversity. These melanoma phenotypes are governed by distinct GRNs and are defined by the expression of specific markers. From the different gene signatures of melanoma states described in several studies, we generated a Venn diagram (Figure 3) that shows the distribution of shared and specific genes of main melanoma states between these publications [5,7,25,43]. Interestingly, one can notice that a high number of genes are conserved among these signatures in line with their main roles as regulators of cancer plasticity and melanoma states. Specific genes can also be identified for each signature but this may be related to the diversity of cell lines and importantly to the culture conditions (microenvironment) used in these studies for gene expression profiling.

Different trajectories have been identified and cells follow along a multistage differentiation model. This phenotypic diversity, also characterized by a highly dynamic cell state transition, is a prominent source of tumor heterogeneity and a key driver of tumor progression and resistance to therapy.

Given its high phenotypic plasticity, melanoma is considered a suitable model to understand the complex relationship between cell phenotype, invasion, and drug resistance. The dynamic changes of differentiation–state transitions well-characterized in melanoma are also observed in other cancers. Furthermore, the transcriptional networks that dictate melanoma phenotypes and drive dedifferentiation are associated with EMT. Therefore, all of the above suggest that this deep characterization of the phenotypic heterogeneity in melanoma will be very important for a better understanding of other cancer types and mechanisms underlying their metastatic processes.

## 3. Role of Epigenetic Modifications in the Dynamic Phenotype Switch

Upon exposure to microenvironment stress signals or drug therapy, melanoma cells switch their phenotype in order to adapt to stress. This phenotypic plasticity is associated with epigenetic modifications that include chromatin remodeling, methylation, and activities of long non-coding RNAs (lncRNA)/miRNAs (Figure 1).

### 3.1. Chromatin Remodeling

Chromatin is a highly organized DNA-protein structure. Dynamic changes in chromatin states are essential for transcriptional regulation that include histone post-translational modifications such as methylation, acetylation, and phosphorylation resulting in transcriptional activation or repression [51].

Changes in the chromatin landscape underlie the proliferative/differentiated and the invasive/undifferentiated states in melanoma. Indeed, open chromatin profiling and chIP-seq against the H3K27ac and H3K27me3 reveal an open and active SOX10 promoter in proliferative/differentiated melanoma cells, while the latter promoter displays the H3K27me3 repressive mark in the invasive/undifferentiated melanoma cells. Furthermore, enhancers active in the proliferative phenotype overlap with those in melanocytes. In contrast, enhancers active in the invasive phenotype overlap with fibroblasts that have a mesenchymal regulatory program (Figure 1). Noticeably, dynamic long-range chromatin interactions play a critical role in transcription activation. For instance, SOX9 interacts at a long distance by enhancer–enhancer and/or enhancer–promoter looping. The loop architectures are different between proliferative and invasive phenotypes [5]. As SOX10 and SOX9 are, respectively, the key regulators of proliferative/differentiated and invasive/undifferentiated phenotypes, the above findings highlight the role of chromatin in modulating transcription states that drive the melanoma phenotype.

Furthermore, JARID1B (KDM5B) is an H3K4me3 histone demethylase implicated in the regulation of melanoma phenotype and metabolome reprogramming. JARID1B/KDM5B is expressed at various stages in melanoma. In previous studies, it was shown that high expression of JARID1B/KDM5D is associated with a slow-cycling cell state [52]. Recently, it was demonstrated that JARID1B promotes a differentiated phenotype. Indeed, Chauvistré et al. 2020 [53] suggested that JARID1B/KDM5D acts as a dynamic coordinator of both differentiation and cell division programs. Thus, high KDM5B expression is associated with a slow-cycling differentiated phenotype that allows melanoma cells to cope with stress and survive under treatment. Likewise, another study revealed that drug-tolerant state is associated with high expression of H3K4 demethylases KDM1B, KDM5A, KDM5B [54]. Altogether, these findings show that microenvironment cues or therapies can induce the upregulation of histone demethylases that reshape the chromatin landscape, thereby regulating melanoma phenotypic plasticity.

The switch/sucrose non-fermentable (SWI/SNF) complex is another effector that regulates the chromatin remodeling processes. This complex is divided into two forms: BRG1/BRM associated factor (BAF)/SWI/SNF-A and polybromo-associated BAF (PBAF), known also as SWI/SNF-B. The SWI/SNF family uses ATP energy to remodel the chromatin structure [55]. In melanoma, PBAF comprising BRG is required for the survival of MITF^high^ proliferative melanoma cells. MITF and SOX10 actively recruit BRG1 to chromatin to establish the epigenetic landscape of the proliferative phenotype [56].

### 3.2. Methylation

DNA methylation is another epigenetic mechanism that regulates the gene expression and thereby phenotypic plasticity in melanoma [57]. Distinct DNA methylation signatures underlying the proliferative/differentiated and the invasive/undifferentiated states were identified. Cheng et al. 2015 showed that proliferative melanoma cells have a lower expression of SOX9 than invasive ones due to promoter hypermethylation (Figure 1) [57].

### 3.3. Long Non-Coding RNAs (lncRNAs)

Long non-coding RNAs (lncRNAs) are a group of endogenous noncoding RNAs defined by their length of more than 200 nucleotides. LncRNAs can modulate gene expression through diverse mechanisms, including chromatin remodeling and transcriptional activation/repression and can act also as enhancers, decoys, and scaffolds [58].

LncRNA components of MITF-SOX10 networks are an important class of melanoma phenotype regulators. One of these, disrupted in renal carcinoma 3 (DIRC3) can regulate chromatin accessibility at their sites of expression. High levels of DIRC3 were found associated with an invasive MITF–SOX10 low phenotype at the primary lesion site. However, at metastatic sites, melanoma may switch back from invasive towards a proliferative state by reestablishing MITF–SOX10 and inhibiting DIRC3 to enable proliferation [59].

In addition, it was shown that high expression of lncRNA ZEB1–AS1 is associated with the invasive phenotype [60]. In contrast, the expression of ZEB1–AS1 is inversely correlated with the proliferative/differentiated signature, thereby promoting an invasive phenotype [60].

Otherwise, several studies have linked various LncRNA with invasion. For example, the lncRNA SNHG7 (small nucleolar RNA host gene 7) promotes invasion in melanoma via the upregulation of SOX4, another master regulator of EMT [61]. Indeed, it was shown that TGF-β-mediated expression of SOX4 can induce EMT in human mammary epithelial cells [62]. The lncRNAs—promoter of CDKN1A antisense DNA damage activated RNA (PANDAR) and SRA-like non-coding RNA1 (SLNCR1)—can also promote melanoma invasion through EMT regulation and upregulating MMP9 [63,64].

Recently, EGR1 was shown to bind lncRNA SLNCR1 to promote melanoma growth and invasion [65]. In addition, EGR3 was identified as a master regulator of the intermediate phenotype. These findings suggest a possible role for the EGR-SLNCR-1 network in the fine-tuned regulation of the intermediate phenotype sharing both the proliferative and invasive features.

Altogether, LncRNAs emerge as new epigenetic regulators of the phenotypic plasticity in melanoma.

### 3.4. miRNAs (MicroRNAs)

MicroRNAs (miRNAs) are a family of endogenous, small non-coding RNAs that regulate gene expression [66,67]. It has been reported that miRNAs are dysregulated in several cancers including melanoma and they are implicated in several stages of carcinogenesis including both initiation and progression [66,67]. miRNAs can regulate melanoma plasticity. For instance, it was shown that miR-410-3p can upregulate AXL and drive a phenotypic switch towards an invasive phenotype in melanoma cells [68]. In addition, miR-182 is another key regulator of the EMT that promotes an invasive phenotype by directly repressing FOXO3 (forkhead box O3) and MITF [69]. On the other hand, miR-211 is inversely correlated with an invasive phenotype in melanoma [66]. Additionally, miR-542 3p inhibits invasion and EMT in preclinical models of melanoma [66]. 

Collectively, these results show that miRNAs have a critical role in the regulation of phenotype switching in melanoma.

## 4. Importance of the Metabolic Rewiring in the Phenotypic Switch

The phenotypic plasticity in melanoma is also driven by metabolic rewiring. Metabolic differences underlie the distinct melanoma states (Figure 1).

### 4.1. Glycolysis and Warburg Effect vs. OXPHOS

The Warburg effect is defined by the preferential use of glycolysis with lactate production for ATP synthesis even under normoxic conditions [70]. This phenomenon is a hallmark of cancer cells, which presents high proliferation rates. Melanoma cells have a highly glycolytic rate to sustain their growth by producing ATP and intermediate metabolites for macromolecule biosynthesis. Thus, the Warburg effect is high in the proliferative melanocytic phenotype and is low in the invasive mesenchymal-like phenotype [71,72]. In contrast, OXPHOS is elevated in cells with an invasive mesenchymal-like phenotype, which are slow-cycling, and is decreased in rapidly dividing cells presenting a melanocytic phenotype [52,73,74].

These metabolic adaptations contribute to tumor plasticity and phenotype switching and confer a survival advantage to melanoma cells. 

Furthermore, regarding sensitivity to treatments, MAPKi indirectly target glycolysis [75]. Accordingly, melanoma cells with a melanocytic phenotype defined by a glycolytic metabolism are sensitive to these inhibitors, while cells with a mesenchymal-like phenotype and mainly using OXPHOS for their energy production are less sensitive. 

### 4.2. Glutamine Metabolism

Glutamine is an abundant amino acid with many roles in tumor progression. Its metabolism has been associated with resistance to targeted therapy [75,76]. In melanoma, it can act as a main driver of melanoma aggressivity. Indeed, it is able to upregulate HIF1α and its key targets, BNIP3 and Twist, to induce a mesenchymal-like switch and promote migration and invasion [77]. As resistance to targeted therapy is associated with the mesenchymal-like phenotype, one may expect that glutamine metabolism plays a major role in cells presenting such phenotype.

### 4.3. Nicotinamide Phosphoribosyltransferase (NAMPT)

Ohanna et al. 2018 [78] showed that NAMPT promotes a phenotypic switch in melanoma towards an invasive state. Nicotinamide phosphoribosyltransferase (NAMPT) is a critical enzyme for NAD^+^ production, an essential cofactor that controls a variety of processes, including energy metabolism. NAMPT can induce the activation of ZEB1 and TGF-β pathways and the downregulation of genes associated with the proliferative state. In their study, the authors showed that NAMPT favors phenotype switching through epigenetic remodeling. Furthermore, a proteomic study reported an upregulation of NAMPT in melanoma cells resistant to MAPKi along with a mesenchymal-like phenotype [79].

### 4.4. Lipid Metabolism

Many recent studies have focused on lipid metabolic reprogramming and its link with melanoma progression [80]. 

MITF is a lineage-specific regulator of the fatty acid desaturase SCD. Stearoyl-CoA desaturase (SCD) is an endoplasmic reticulum (ER)-bound lipogenic enzyme critical for the biosynthesis of monounsaturated fatty acids (Fas) [81]. SCD is associated with a MITF^high^ proliferative phenotype. Saturated fatty acids (SFAs) and monounsaturated fatty acids (MUFAs) provide an important source of energy required for melanoma proliferation. Thus, proliferative melanoma cells display high activity of enzymes involved in fatty-acid biosynthesis, including SCD. On the other hand, low SCD activity is associated with an invasive phenotype (NF-kB/ATF4/AXL^high^ ). Therefore, the MITF-SCD axis seems to play an important role in regulating melanoma plasticity. 

Furthermore, it was demonstrated by Leclerc et al. 2019 [82] that lysosomal acid ceramidase ASAH1, a key enzyme of sphingolipid metabolism, controls phenotype switching in melanoma. Melanoma cells with high expression of ASAH1 display a proliferative state whereas low ASAH1 expression is associated with an invasive phenotype mediated by the activation of integrin/FAK signaling cascade. Indeed, ASAH1 was identified as a MITF target, thereby implying MITF in the regulation of sphingolipid metabolism. In addition, MITF controls the TCA cycle [83] and regulates PGC1α, the master regulator of oxidative metabolism [84]. 

Altogether, these findings suggest that MITF can also dictate the melanoma phenotype through lipid metabolism regulation.

## 5. Prominent Role of Microenvironment Stress Signals in Phenotype Switching

Melanoma shows high phenotypic heterogeneity with distinct subpopulations that switch dynamically in response to different microenvironmental cues, mainly hypoxia, inflammation, and nutrient deprivation (Figure 4).

### 5.1. Hypoxia

It has been well established that hypoxia plays an important role in regulating phenotype switching in melanoma towards an invasive/undifferentiated phenotype [26,85]. Indeed, hypoxia leads to MITF repression via a mechanism involving the transcription factor bHLHE40/DEC1 [86]. Furthermore, HIF1α and HIF2α can drive melanoma invasion through PDGFRα and FAK-mediated activation of SRC [87]. Moreover, hypoxia guides a switch from ROR1 to ROR2, which is critical for Wnt5A-mediated invasion and metastasis [26].

On the other hand, recent findings by Louphrasitthiphol et al. 2019 [83] suggest that early response to hypoxia can include a transient upregulation of MITF by HIF1α and that MITF coregulates a set of hypoxia response genes, including VEGF-A and SLC5A9. In the same study, the authors show that MITF can repress its own expression by a negative feedback loop and that melanoma cells with distinct phenotypes display different responses to hypoxia.

Altogether, the above findings suggest that the crosstalk between hypoxia and MITF plays an important role in phenotype switching and that cells within tumors can adapt differently to changes in oxygen availability.

### 5.2. Inflammation

The interplay between inflammation and phenotype switching in melanoma has also been addressed in several studies [22,88]. The pro-inflammatory TNFα can promote an interconversion among melanoma phenotypes. Inflammation-induced plasticity contributed to immune escape and resistance to adoptive T cell therapy in melanoma mice models. Indeed, TNFα induced dedifferentiation of melanoma cells and abrogated immune recognition by adoptively transferred pmel-1 [22,88].

It was also identified that the antagonism between MITF and c-Jun acts as a molecular interface between pro-inflammatory signals from the TME and melanoma plasticity. Indeed, MITF negatively regulates c-Jun expression through direct binding to c-Jun enhancer. MITF downregulation, in response to pro-inflammatory signals, stimulates c-Jun expression, which in turn amplifies TNF-stimulated cytokine expression, leading to further MITF suppression. This molecular cascade promotes an inflammatory MITF low/c-Jun high cell state favoring myeloid cell recruitment [22,88].

All of the above findings show that inflammation promotes phenotype switching towards an undifferentiated/invasive state associated with a myeloid cell-rich tumor microenvironment.

### 5.3. Nutrient Deprivation

An essential role for glutamine deprivation in melanoma was suggested [89] to suppress MITF, promote invasion, and upregulate genes associated with EMT. Glutamine deprivation also favors a translation reprogramming via eIF2B inhibition and activating transcription factor 4 (ATF4) expression that is an evolutionarily conserved response to cope with stress [89].

Glucose restriction promotes ROS production, which also upregulates ATF4, leading to an invasive MITF-low phenotype [90].

Along with nutrient limitation, microenvironmental cues, such as inflammation, converge to inhibit eIF2B by promoting eIF2α phosphorylation, which represses MITF and activates activating transcription factor 4 (ATF4), a key mediator of the nutrient-sensing response pathway (Figure 4) [89]. Interestingly, inhibition of eIF2B and the consequent translation reprogramming promotes tumor colonization, resistance to T-adoptive therapy, and correlates with the gene expression signature that predicts poor response to anti-PD-1 immunotherapy (innate anti-PD-1 resistance (IPRES) gene expression signature) [89].

Therefore, these findings show that the ATF4–eIF2 axis is crucial in regulating cell plasticity in response to multiple microenvironmental signals. This translation reprogramming is a key determinant of melanoma invasion and resistance. 

In line with these findings, it was demonstrated that glutamine deprivation induces translation reprogramming through mTOR and eIF2 signaling and promotes migration in a model of breast cellular transformation [91]. They also showed in this study the interconnection between inflammation and translation reprogramming [91].

Translation reprogramming is controlled via two key cis motifs: upstream open reading frame (uORF) and internal ribosome entry site (IRES) sequences. The uORFs are found in genes induced by stress (e.g., ATF4) and inhibit the translation of downstream main open reading frames in normal growth conditions [92]. On the other hand, IRESs are highly structured motifs predominantly located in the 5′UTR of stress response genes and drive translation in suboptimal growth conditions [92]. uORF and IRES-dependent translation are regulated by mTOR and EI2F signaling. All of these mechanisms are triggered by cells in response to stressful stimuli [92]. 

Along with uORF and IRES, DEAD-box helicase 3 X-linked (DDX3X) was also shown to impact global protein synthesis through its effects on translation initiation [93]. DDX3X can dictate translation reprogramming and metastasis in melanoma. Indeed, DDX3X controls MITF mRNA translation via IRES. Dysfunction of DDX3X leads to the selection of highly invasive MITF-low-expressing cancer cells. Moreover, reduced DDX3X expression correlates with tumor progression and poor prognosis in melanoma patient cohorts [93].

In conclusion, the translation reprogramming is a conserved mechanism in cancer progression adopted by cancer cells to survive under stress conditions including nutrient limitation, inflammation, low oxygen, and exposure to therapy.

### 5.4. Tumor Microenvironment (TME) Factors

#### 5.4.1. Cancer-Associated Fibroblasts (CAFs)

Cancer-associated fibroblasts (CAFs) are the most dominant components in the tumor stroma. They are involved in several processes, including matrix remodeling, growth factors, and cytokine production [94]. CAF-derived TGF-β induces an invasive MITF^low^/AXL^high^ phenotype in melanoma cells [95]. Therefore, the reciprocal interactions between melanoma and CAFs can promote melanoma cell plasticity. 

#### 5.4.2. CD73

CD73 is an ectoenzyme essential for the production of extracellular adenosine that promotes an immunosuppressive tumor microenvironment. Indeed, adenosine inhibits T helper type 1 (Th1), the cytotoxic activity of natural killer (NK) cells, and macrophage activation. Moreover, adenosine mediates angiogenesis by inducing the production of VEGF [96]. 

The induction of CD73 is linked to phenotypic plasticity. Indeed, CD73 expression is associated with the invasive state [97]. CD73 expression is stimulated by MAPK signaling and TNF-α through the c-Jun/AP-1 transcription factor complex [97] or by hypoxic conditions [98]. Thus, CD73 emerges as a new central regulator of stress responses and, thereby, melanoma plasticity.

#### 5.4.3. Interleukin-like EMT Inducer (ILEI)/FAM3C

The family with sequence similarity 3 (FAM3) cytokine-like gene consists of four members (FAM3 A, B, C, and D). FAM3C, called interleukin-like EMT inducer (ILEI), is a secreted factor that is involved in TGF-β-mediated EMT and has been found to promote metastasis in several types of cancer [99]. 

ILEI (FAM3C) mRNA is highly expressed in melanoma metastases. It was demonstrated that cells with proliferative MITF^high^ phenotype display low levels of ILEI (FAM3C) while cells with invasive MITF^low^ phenotype show high levels of ILEI [100]. Noteworthy, the phenotype switching is linked to an upregulation of ILEI (FAM3C). ILEI-regulated genes are enriched for JUN signaling a central node in cellular stress signaling and regulator of the invasive phenotype [100].

These findings reveal an important role for ILEI (FAM3C) as a novel secreted factor regulating melanoma invasiveness.

#### 5.4.4. Endothelin-1 (ET-1)

The endothelins (ETs) comprise a family of three peptides: ET-1, ET-2, and ET-3. ETs regulate several processes in cancer, including cell survival and cell invasion through the binding to distinct G protein-coupled receptors (GPCR): ETAR (EDNRA), and ETBR (EDNRB) [101].

ETs (EDN) induce the activation of MMPs (MMP2 and MMP9) [101]. Of note, the crosstalk between hypoxia, endothelial, and melanoma cells promotes melanoma invasiveness and regulates vascularization via ET-1. Indeed, ET-1 is upregulated under hypoxic conditions, which in turn induce VEGF-A and VEGF-C secretion [102]. 

A recent study showed that ET-1 expression is upregulated under treatment with MAPK inhibitors. More important, ET-1 (EDN1) supports MITF^high^ melanoma cells via endothelin receptor B (EDNRB) and AXL^high^ through EDNRA [103].

Thus ET-1 contributes to the promotion of an invasive phenotype and, more importantly, seems to be a critical regulator for the maintenance of phenotypic heterogeneity.

## 6. Critical Role of Oxidative Stress in Regulating Phenotypic Plasticity

Reactive oxygen species (ROS) have been implicated in all aspects of melanoma development and play a critical role in regulating cell survival and cell invasion. Further, melanoma cells encounter several microenvironmental cues, such as hypoxia, glucose deprivation, and inflammation, which all promote oxidative stress. Thus, ROS levels can regulate melanoma phenotype by acting as key mediators between microenvironment signals and effectors of pathways that govern genome and epigenome, allowing cell adaptation.

### 6.1. ROS Sources

In melanoma cells, ROS is mainly generated by melanosomes/melanogenesis, the NADPH oxidase family of proteins (NOX), and mitochondria (Figure 5).

#### 6.1.1. Tyrosine-Induced Melanogenesis

Melanoma cells are well known for their ability to produce melanin. This process, called melanogenesis or pigmentation, is the main function of the melanocyte and is preserved after its malignant transformation. Melanin synthesis occurs in specialized intracellular organelles called melanosomes. Tyrosine is the precursor of melanin synthesis and the substrate of tyrosinase the key enzyme of melanogenesis [104]. Melanin is derived from “tyrosine” through a sequence of oxidation reactions involving superoxide anion (O_2_−) and hydrogen peroxide (H_2_O_2_) generation [104]. Thus, melanogenesis stimulation can promote oxidative stress and alters mitochondrial respiration [105,106]. It was shown that MAPK inhibitors [107] and several paracrine factors upregulated in melanoma, such as endothelin and POMC-derived peptides, can promote melanogenesis [103,108].

In a recent study, we investigated the role of tyrosine as a stimulator of melanogenesis and, hence, oxidative stress on melanoma plasticity. Indeed, we showed that tyrosine-induced pigmentation can rapidly promote a phenotypic switch towards either a mesenchymal-like or a senescence-like state. This phenotypic plasticity is an adaptive mechanism that allows cells to cope with a continuous stress as long as tyrosine stimulates melanogenesis [50].

The above findings show the importance of tyrosine and ROS in regulating melanoma phenotypic plasticity.

#### 6.1.2. NADPH Oxidases (NOX)

The NADPH oxidases (NOX) family is composed of seven members, including NOX1–5, Duox1 and Duox2. These membrane-bound proteins generate cytosolic ROS and upregulate EMT markers, such as MMP2 and MMP9 and, thus, contribute to melanoma invasion [109,110].

NOX-derived ROS promotes invasion in pancreatic cancer, colon cancer, and melanoma by promoting EMT. For instance, NOX4 is induced by hypoxia and can induce histone modifications that activate TGF-β and/or SNAIL, a key EMT transcription factor [111]. 

We also showed that phenotype switching is associated with ROS production and NOX activation in melanoma [50].

Thus, NOX is an important source of ROS and may have a critical role in the ROS-mediated invasive melanoma phenotype. 

#### 6.1.3. Mitochondria

Mitochondria is also involved in the production of ROS through oxidative phosphorylation along the electron transport chain [112]. The MITF–PGC1α axis controls the mitochondrial energy metabolism and also improves the ROS scavenging capacity which enables survival under oxidative stress conditions [96]. Thus, mitochondria activity may be of prime importance in the mechanisms of sensitivity or resistance to targeted therapies.

### 6.2. MITF-PGC1α Axis in the Regulation of Oxidative Stress and Phenotypic Plasticity

The peroxisome proliferator-activated receptor-γ coactivator (PGC1) PPARGC1A family is composed of three members: PGC1α, PGC1β, and PPRC1, the best-studied and well known being PGC1α [113]. 

PGC1α is a master regulator of mitochondrial biogenesis and function. It plays also a central role in ROS detoxification by regulating the expression of several antioxidant genes including thioredoxin (TRX), thioredoxin reductase (TRXR), manganese superoxide dismutase (MnSOD/SOD2), and catalase (CAT) [114].

The melanocyte lineage factor MITF directly drives the expression of PGC1α and thereby mitochondrial respiration [84,113]. Melanoma cells with melanocytic phenotype display high MITF and PGC1α expression, increased mitochondrial energy metabolism, and ROS detoxification capacities that allow cells to cope with oxidative stress [84,113]. On the other hand, less differentiated melanoma cells have low MITF and PGC1α expression, are more glycolytic, and exhibit high ROS production [84,113]. Thus, PGC1α levels have an important impact on the metabolic state, and melanoma cells can switch through different phenotypes during cancer progression with alternate high or low expressions of PGC1α.

Using the interactive web-interface resource provided by Tsoi et al. 2018 [25], we confirmed the correlation between MITF and PGC1α expression in the panel of melanoma lines used. [25]. Noteworthy, we found that PGC1α is highly expressed in the melanocytic (C4) and the transitory (C3) phenotypes whereas this expression is low in the invasive phenotypes (neural crest-like and undifferentiated phenotypes) (Figure 6). Of note, by re-examining our previous microarray data [115], we found that PCG1α was also significantly correlated with a panel of genes coding for enzymes or tyrosine transporters involved in melanogenesis (TRYP1, DCT, OCA2, TYR, MITF) and, thus, associated with a given melanocytic phenotype.

Altogether, these findings can be explained by the fact that PGC1α can act as a potent suppressor of the invasive phenotype by downregulating multiples genes within the TGF-β/WNT pathways [114]. Moreover, Luo et al. 2016 demonstrated that PGC1α can suppress the invasive phenotype in melanoma by acting on the ID2–TCF4–integrin axis [116]. Mechanistically, PGC1α directly increases transcription of ID2, which in turn binds to and inactivates the transcription factor TCF4 an important modulator of metastasis and EMT. A recent study by Luo et al. 2020 [117] showed that PGC1α expression can be silenced by an epigenetic modification that involves histone H3K27 trimethylation. H3K27me3-mediated PGC1α gene silencing promotes melanoma invasive phenotype through TCF12/WNT5A/YAP axis [117].

Comparably to MITF, inflammation leads to the down regulation of PGC1α [118]. Moreover, PGC1α and hypoxia-inducible factor 1α (HIF1α) are inversely correlated [119]. Thus, these microenvironmental cues can promote oxidative stress via the downregulation of PGC1α the powerful mediator of ROS removal.

Collectively, these findings reveal the importance of the MITF-PGC1α axis in the regulation of oxidative stress and thereby phenotypic plasticity.

### 6.3. Oxidative Stress Defines the Invasive/Undifferentiated Phenotype

Oxidative stress can regulate phenotypic plasticity in melanoma. For instance, glucose restriction results in an increase of ROS levels and leads to ATF4-mediated MITF suppression [90]. Moreover, it was shown that fibroblasts within the aged microenvironment can induce ROS in melanoma cells which contributes to melanoma progression [120]. Furthermore, a report suggests that there is a significant relationship among circulating TGF-β1 levels, systemic oxidative stress, and metastatic spread in melanoma patients [121].

Moreover, Tsoi et al. 2018 [25] identified high levels of ROS in undifferentiated/invasive melanoma lines along with lower basal levels of reduced glutathione that is associated with an increased sensitivity to ferroptosis induction [25].

Otherwise, oxidative stress is also implicated in the resistance to targeted therapy in melanoma. Indeed, the resistance to MAPKi targeted agents is associated with elevated ROS levels [25]. GSH levels are also downregulated in MAPKi-resistant melanoma lines. Likewise, antioxidant genes are decreased in drug-tolerant persister cancer cells as well, which constitute an important reservoir for the emergence of acquired resistance leading to relapses [122]. In line with these findings, we identified ROS as a driver of tyrosine-induced phenotypic switching [50]

Based on the above, there is compelling evidence that oxidative stress is a key determinant of the undifferentiated state and acquired resistance to therapies not only in melanoma but also in other cancer types.

## 7. Targeting Phenotypic Plasticity: Identifying New Vulnerabilities of Melanoma Phenotypes and Future Challenges

A deep understanding of the distinct melanoma phenotypes, their key regulators, the associated metabolic pathways, and the microenvironment signals is critical to identify and explore new vulnerabilities, which can constitute a very attractive strategy to overcome therapy resistance (Figure 7 and Table 1).

### 7.1. Targeting Phenotype-Specific Vulnerabilities 

#### 7.1.1. Targeting AXL Receptor Tyrosine Kinase as a New Promising Approach

Axl is a member of the Tyro3, Axl, Mer (TAM) subfamily of receptor tyrosine kinases (RTKs). AXL is overexpressed in several cancer types and correlates with poor survival. AXL is activated by GAS6 ligand and can promote cell invasion, EMT, and angiogenesis. Moreover, AXL regulates the immune response and is implicated in the stem cell maintenance. More important, AXL contributes to drug resistance in cancer treatment [123].

In melanoma, AXL expression is associated with an invasive phenotype and thereby resistance to MAPKi. Thus, it represents an important target to inhibit this phenotype. Accordingly, it was demonstrated that the combination of AXL-107-MMAE with MAPKi cooperate to inhibit melanoma growth in vitro and in vivo [23]. AXL-107-MMAE is a specific AXL-targeted antibody–drug conjugate (ADC) generated by conjugating the human antibody with the microtubule disrupting agent monomethyl auristatin E (MMAE). AXL-107-MMAE is safe and tolerable and its efficacy was investigated in several tumor types [23]. Recently, Boshuizen al. 2021 showed that targeting immunotherapy-resistant melanoma and lung cancer using an AXL-targeting ADC enhances the sensitivity to immune checkpoint inhibitors (IC) [142]. 

Small molecule AXL inhibitors have also been proposed as new therapeutic strategies in melanoma. Indeed, bemcentinib is an oral, highly selective AXL inhibitor that shows promising results in preclinical and clinical studies, both alone and in combination, mainly with immune checkpoint inhibitors in multiple solid and hematologic tumors [124,125]. For instance, bemcentinib was investigated in combination with either dabrafenib/trametinib (D/T) or IC pembrolizumab in melanoma patients [125]. Dubermatinib is another selective AXL inhibitor that was investigated in CLL (NCT03572634) and solid tumors (NCT02729298). Other non-selective AXL inhibitors that have the latter among their targets, such as crizotinib, bosutinib, and cabozantinib, and multi-targeted TKI, such as sunitinib, foretinib, and sitravatinib, are also available in clinics [126]. 

Altogether, these findings show that targeting AXL can be a central approach to suppress the invasive phenotype in melanoma.

#### 7.1.2. Targeting The Neural Crest Stem Cell (NCSC) State and RXRG Signaling to Delay Melanoma Relapse 

In melanoma, the relapse following treatment with MAPKi is driven by a small subpopulation of residual or “drug-tolerant” cells, termed minimal residual disease (MRD). MRD can contain four distinct drug-tolerant transcriptional states [43]. One of these states is characterized by high expression of neural crest stem cell (NCSC) markers. This NCSC subpopulation has a critical role in driving relapse.

The neural crest (NC) is a multipotent stem cell population generated during early vertebrate. NCSCs arise at the developing dorsal neural tube and have high migratory capacity [143]. These cells disseminate into the whole embryo to differentiate into a variety of cell types with different functions, such as peripheral nervous system cells, bone, cartilage, connective tissue, endocrine cells, and pigment cells. The acquisition of invasive phenotype and resistance is associated with gene regulatory networks reminiscent of this neural crest [143].

The retinoid X receptor gamma (RXRG) is the key driver of the NCSC state. Targeting the NCSC subpopulation using pan-RXR antagonist (HX531) enhances the antitumor activity of MAPKi, blocks the emergence of NCSCs, and prevents disease relapse of patient-derived melanoma [43]. Noteworthy, the pharmacological inhibition of RXRG lead to a concomitant increase in the invasive AXL-high subpopulation. Blocking the latter subpopulation is achievable and described in the section above.

All of the above findings illustrate the remarkable heterogeneity in melanoma and suggest that inhibition of these different invasive phenotypes is a very promising approach to overcome resistance and prevent relapse.

#### 7.1.3. Targeting Ferroptosis to Block the Dedifferentiation Resistance Escape Route

Ferroptosis is a type of programmed cell death that occurs via an iron-dependent accumulation of lipids peroxides [144]. It is regulated by glutathione peroxidase 4 (GPX4). GPX4 is a selenoprotein that converts reduced glutathione (GSH) into oxidized glutathione (GSSG) and thereby reduces lipid peroxides (L-OOH). Inhibition of GPX4 activity leads to the accumulation of cytotoxic lipid peroxides, a hallmark of ferroptosis [144].

It was reported that the EMT-like or persister cell state in cancer cells is highly dependent on GPX4 activity for survival [122]. Accordingly, Tsoi et al. 2018 [25], in their study, showed that the undifferentiated phenotype frequently linked to innate and acquired resistance to targeted and immune therapies in melanoma is associated with increased sensitivity to ferroptosis inducing compounds, including erastin, ML162, and ML21s. Whereas the undifferentiated phenotype displays high sensitivity, the melanocytic state is the most resistant one.

Therefore, these findings support the rationale for combination strategies using ferroptosis-inducing drugs to block the dedifferentiation resistance escape route. Interestingly, there is new evidence demonstrating that ferroptosis-inducing therapy can enhance the effect of immune checkpoint inhibitors anti-PDL1 [144].

#### 7.1.4. Targeting the Senescence-like Phenotype

The senescence-like phenotype is a form of tumor dormancy present in a subpopulation of tumor residual cells that escaped cell death following treatment [47]. Indeed, the senescence-like cells display resistance to apoptosis and secrete cytokines and growth factors referred as senescence-associated secretory phenotype (SASP) [47]. This SASP is associated with angiogenesis, invasion, and can promote metastasis [47]. We previously showed that MAPKi can promote a senescence-like phenotype in melanoma cells [49]. Furthermore, we identified that stressful microenvironmental conditions with high tyrosine induce a phenotypic switch towards an invasive or senescence-like state in melanoma primary cultures [50]. Altogether, these findings show that targeting the senescence-like phenotype is considered as an intriguing therapeutic strategy. Senolytics are a class of drugs that selectively eliminate senescent cells. These drugs include inhibitors of Bcl2 family, histone deacetylase inhibitors, inhibitors of PI3K/AKT/mTOR pathway, TKI, and flavonoids [127,128]. 

Bcl2 inhibitors, such as the BH3 mimetic navitoclax and the HDACi panobinostat, promote senescent cell apoptosis and are able to eliminate senescent cells that persist during chemotherapy [127,128]. Likewise, the combination of dasatinib and quercetin (Q + C) is also very promising in the clearance of senescent cells [127,128]. Navitoclax and the combination Q + C can also inhibit the SASP and reduce inflammation [127,128]. Moreover, the PI3K/AKT pathway is a main survival pathway in senescent cells. Thus, blocking the latter can be very promising, since PI3K/AKT inhibitors are investigated in clinic in several cancers [127,128].

Collectively, developing effective approaches to eliminate residual senescent-like cells in melanoma can be very promising to prevent relapse and inhibit metastasis. Nevertheless, exploring the best strategy and the optimal schedule of treatment is needed in future clinical applications.

#### 7.1.5. Targeting Endothelin Receptor Signaling as a Unique Approach to Overcome Phenotypic Heterogeneity 

The phenotypic heterogeneity is a key driver for therapy failure leading to the establishment of tumor states associated with acquired resistance and high metastatic potential.

Endothelin 1 (EDN1), which is implicated in several aspects of tumor progression [101], has been identified as a master regulator of phenotypic heterogeneity and can contribute to paracrine protection against MAPKi [103]. Indeed, EDN1 expression is upregulated under treatment and confers resistance through ERK reactivation. More important, EDN1 expression regulated by MITF can protect the proliferative phenotype characterized by MITF^high^ via the endothelin receptor B (EDNRB), and it is crucial for the maintenance of the AXL^high^ invasive phenotype via EDNRA [103].

The combination of EDNR antagonists, such as bosentan, macitentan, and BQ788, with MAPKi, lead to significant tumor growth and improve their response [103]. Targeting EDNR signaling not only inhibits distinct phenotypes but can also suppress a favorable microenvironment for tumor growth and progression through acting on cancer associated fibroblast and endothelial cells [101,103].

Therefore, targeting endothelin receptors can be a novel approach in melanoma for tackling phenotypic heterogeneity.

### 7.2. Targeting Lipid Metabolism as a New Strategy to Overcome Phenotypic Plasticity

Lipid metabolism contributes to melanoma plasticity and aggressiveness. Indeed, it regulates the dedifferentiation process and sustains cell growth and metastasis in melanoma [81,129]. Thus, targeting the lipid metabolic network emerges as an exciting new approach in cancer research and drug discovery. 

Fatty acid (FA) synthesis is highly active in cancer cells and serves as a fuel source of energy that promotes tumor growth. Most of the enzymes associated with fatty acid synthesis (e.g., FASN, SCD, ACC, ACLY), and receptors related to lipid uptake (e.g., FAT/CD36, FATPs, and FABPpm) are upregulated in several cancers, including melanoma [81,129].

#### 7.2.1. Blocking Fatty Acid Translocase (FAT/CD36)

CD36 is a transmembrane glycoprotein (also known as fatty acid translocase (FAT)) that has an essential role in FA uptake and is involved in metastasis and cancer cell growth [81,129]. In melanoma, the starved melanoma cell (SMC) state found early in minimal residual disease (MRD) is characterized by high expression of CD36 [43]. Accordingly, targeting metastasis-initiating cells by an anti-CD36 antibody showed high efficacy in oral cancer mouse models of human oral carcinomas [129].

Together, these findings show that blocking CD36 might be a promising strategy to prevent metastases and relapse in melanoma.

#### 7.2.2. SCD Modulation

Stearoyl CoA desaturase 1 (SCD1) is an endoplasmic reticulum-associated lipogenic enzyme that converts saturated fatty acids (SFAs) into monounsaturated fatty acids (MUFA) [81,145]. The proportion of SFAs and MUFAs is involved in controlling cell proliferation and differentiation. Likewise, several studies reported an upregulation of SCD in several cancers and its implication in cell proliferation, migration, and metastasis during both early states and progression [81,145]. Therefore, SCD is considered a novel potential target in cancer therapy. 

In melanoma, a strong correlation was found between SCD1 expression and the proliferative phenotype [81]. Vivas-García et al. 2020 showed that MITF^high^ phenotype displays high sensitivity to SCD inhibitor consistent with a strong decrease in expression of genes implicated in cell cycle progression (E2F and MYC targets), whereas invasive/undifferentiated MITF^low^ phenotypes are insensitive to SCDi [81]. This undifferentiated state and thereby the metastatic capacity can be induced under SCDi as well as CD36 expression. Thus, MITF^low^ state may be more dependent on CD36 activity for the uptake of FAs from the microenvironment.

Altogether, these observations suggest that a therapeutic strategy that combines SCD inhibition and CD36 blockade would be very effective to refrain phenotypic plasticity in melanoma.

#### 7.2.3. SREBPs Inhibition

Sterol regulatory element-binding protein-1 (SREBP-1) is a master regulator transcription factor of lipogenesis that regulates the expression of genes involved in lipid synthesis and uptake [80]. SREBP1 has an important role in cancer progression and metastasis. Several small-molecule inhibitors of SREBPs are available, such as fatostatin, botulin, and PF-429242. These inhibitors have demonstrated anti-tumor effects in various cancers in preclinical studies [130]. Importantly, SREBP-1 inhibition can sensitize resistant mutant BRAF melanoma both in vitro and in vivo models [146]. 

These findings suggest that targeting SREBP-1 can offer a new alternative to overcome acquired resistance in melanoma.

#### 7.2.4. Fatty Acid Synthase (FASN) Inhibition

FASN is a central modulator of lipid metabolism catalyzing the last step in de novo-lipogenesis. It also plays a critical role in tumor growth and emerges as a target in cancer [131,132].

The early-generation FASN inhibitors, such as cerulenin and C75, have shown promising anticancer activity, but their pharmacological and off-target effects have limited their clinical development [131,132]. The recently developed TVB-2640 is a more specific inhibitor that is currently investigated in clinical trials in patients with solid tumors in combination with targeted therapy or chemotherapy. Cells with high lipogenic activity seem more sensitive to FASNi TVB-2640, so it can serve as selection criteria for patients that may likely benefit from FASNi therapy [131,132].

### 7.3. Targeting Glutamine Metabolism to Reduce Melanoma Plasticity and Aggressiveness

As glutamine metabolism develops in melanoma cells with acquired resistance to targeted therapies [76], it is believed that melanoma patients could benefit from treatment inhibiting glutamine entry into the cells or its consumption by the cells. Therefore, treatment combinations should be the best way to further induce cancer cell apoptosis.

#### 7.3.1. Blocking Glutamine Import

Glutamine predominantly enters into the cells through the SLC1A5 transporter. Gamma-1-glutamyl-p-nitroanilide (GPNA) is a pharmacological inhibitor of SLC1A1, which has been synthetized [147] to block glutamine uptake. This inhibitor prevents tumor growth and induces apoptosis in lung cancer cells [148], but the GPNA effect has not been evaluated in melanoma yet. Another inhibitor of SLC1A1 is the benzyl serine, which has antiproliferative properties in melanoma cell lines [149].

#### 7.3.2. Blocking Glutamine Use

BPTES (bis-2-(5-phenylacetamido-1,2,4-thiadiazol-2-yl) ethyl sulfide) is a molecule inhibiting the two GLS1 isoforms (GAC and KGA) but not the GLS2. BPTES induces a conformational change to form inactive tetramers. It has been used in vivo and in vitro in a model of BRAF mutated melanoma and reduces tumor growth [133]. However, this inhibitor cannot be used in clinic because of its low solubility in water. CB-839 is a new allosteric inhibitor of GLS1. This molecule has already shows antitumoral activities, both in vitro and in vivo in many cancer types, including melanoma [134]. This inhibitor can be used in clinic and is orally bioavailable. It is currently used in 22 phase I and phase II clinical trials, including one in melanoma (NCT02771626).

### 7.4. Targeting TME Components and Signals

#### 7.4.1. Targeting Cancer-Associated Fibroblasts (CAFs) to Disrupt Melanoma Plasticity

Cancer-associated fibroblasts (CAFs) are critical components of the TME and are implicated in the regulation of cancer progression, immune crosstalk, and phenotypic heterogeneity and plasticity. Targeting cancer cell–CAF signaling interactions and proteins expressed by CAF are considered very attractive strategies [94,150]. 

CAF-secreted factors that promote phenotype switching in melanoma include TGF-β and FGFs. New anti-TGF-β agents have been developed and tested in clinical trials, including monoclonal antibodies, such as fresolimumab in combination with other therapies. These agents show encouraging efficacy and manageable tolerability [137]. 

The FGF/FGFR axis is also a promising therapeutic target in melanoma [151]. Several small-molecule inhibitors of FGFR receptors and other RTK (such as dovitinib) are available in clinics [138].

Moreover, fibroblast activation protein (FAP) is a proline selective serine protease that is overexpressed in various cancers and associated with worse prognosis [135]. FAP-targeted theranostics are small-molecule FAP inhibitors coupled with radiotracers for both diagnostic and therapeutic settings. Thus, FAPIs are very promising tracers for theranostic applications [135]. These molecules showed anti-cancer activity in preclinical studies [135,136] and they are investigated in several clinical trials [152].

Altogether, these findings show that CAF-targeted agents in combination with existing therapies could yield greater benefit in melanoma. 

#### 7.4.2. Targeting CD73 as a New Potential Therapeutic Opportunity 

CD73 emerged as a regulator of melanoma plasticity [97] and can be considered an ideal target of cancer therapy. Indeed, CD73 is an immune inhibitory molecule that is upregulated in cancer and promotes metastases [153].

The safety and efficacy of multiple anti-CD73 monoclonal antibodies, such as oleclumab, CPI-006, and BMS-986179, are under investigation in clinical trials in combination with other therapies in multiple cancers [139]. Anti-CD73 also showed very promising results in melanoma, both in preclinical and clinical studies [140,141].

## 8. Conclusions

Despite recent advances in cancer treatment with the development of targeted therapies and immune checkpoint inhibitors, intratumoral heterogeneity and acquired resistance remain the major limitations. The development of resistance can be driven by both genetic and non-genetic mechanisms. Genetic mechanisms involve the emergence of preexisting clones containing specific genetic mutations, or the acquisition of de novo mutations. On the other hand, non-genetic mechanisms are mainly driven by phenotypic plasticity. In that regard, the natural selection theory (classical Darwinian selection) associated with somatic mutations and the Lamarckian induction concept associated with epigenetic and transcriptional reprogramming, contribute concurrently to phenotypic diversification and, thereby, drug resistance and metastases. In addition, the dynamic and the complex bidirectional interplay between cancer cells and the tumor microenvironment (TME) can govern tumor evolution. ROS can act as a critical mediator between TME signals and effectors of pathways regulating cancer phenotype. In melanoma, the significant advances in single cell RNA techniques and deep learning technologies allow for a detailed characterization and tracing of dynamic phenotypic changes and the identification of different trajectories followed by melanoma cells. Here, our study deepens our understanding of phenotype switching in melanoma and its master regulators. These regulators are involved in cell fate determination, not only in melanoma, but also in several cancer types. This switch towards adaptive phenotypes is orchestrated by the cooperativity of transcriptional, epigenetic, and metabolic networks. The increased knowledge of melanoma plasticity can provide crucial insight into many cancer types and better guide designs for novel therapeutic strategies. Here, we proposed several rational treatment combinations having high potential clinical relevance that aim to target phenotype-specific vulnerabilities, including tumor microenvironments, with the aim to prevent further metastatic dissemination, overcome resistance and, therefore, improve patient outcome.

## Figures and Tables

**Figure 1 cells-11-01157-f001:**
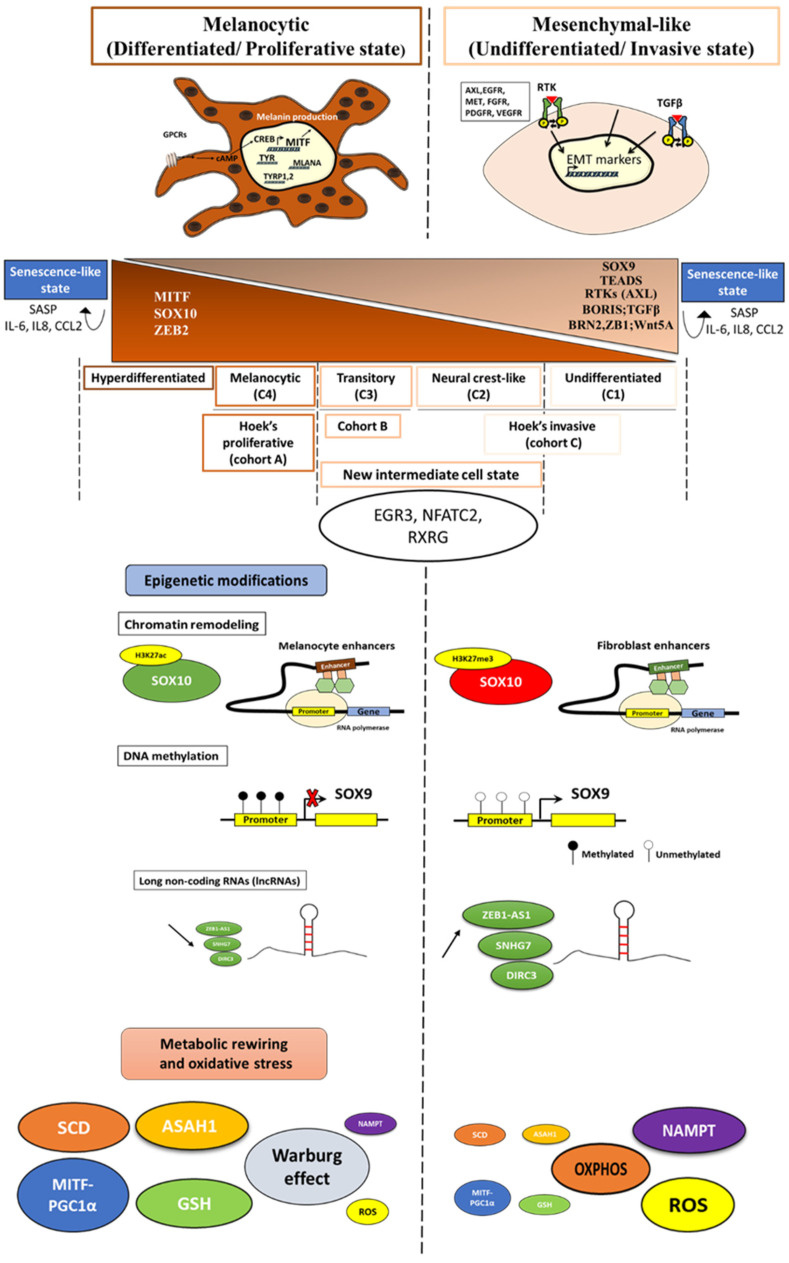
**Schematic description of the complex spectrum of phenotype switching in melanoma.** The distinct melanoma states and their phenotype specific markers, the epigenetic characteristics, the metabolic reprogramming, and the oxidative stress status among the main melanoma phenotypes.

**Figure 2 cells-11-01157-f002:**
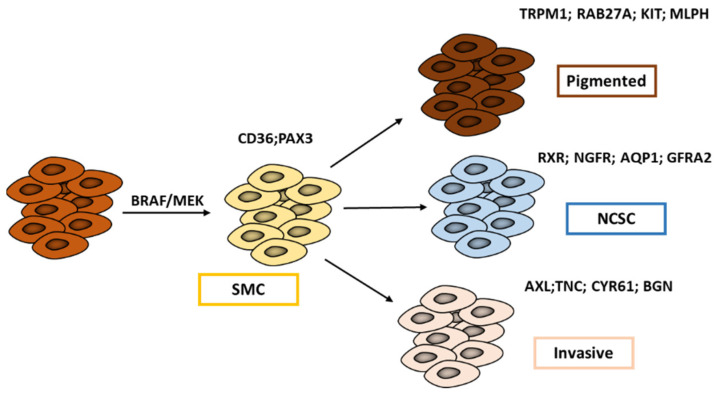
**Melanoma cell states associated with minimal residual disease (MRD).** Exposure to MAPK inhibitors promotes a phenotypic switch from the melanocytic state towards a starved-like state (SMC) facilitating a transition towards either a pigmented/differentiated phenotype or to an undifferentiated one: the classical invasive state or the neural crest stem-like cell (NCSC) state (adapted from Rambow et al. 2018 [43]).

**Figure 3 cells-11-01157-f003:**
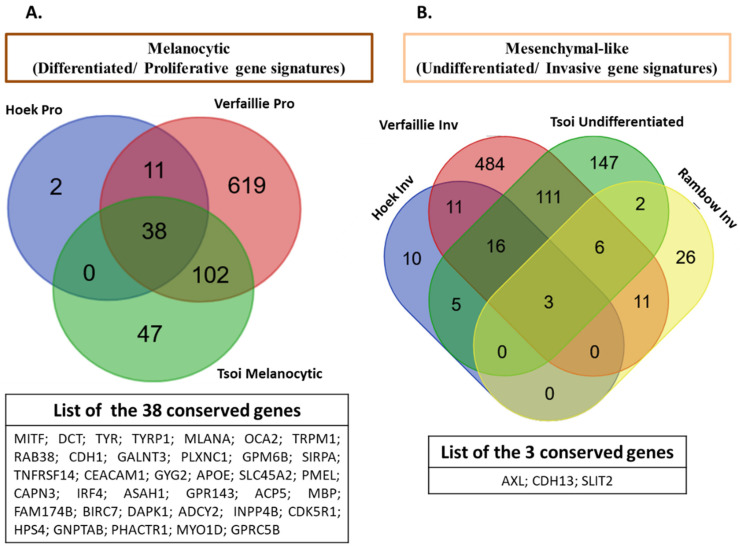
**Comparison of melanoma signatures across different publications.** Venn diagram (http://bioinformatics.psb.ugent.be/webtools/Venn/) showing the distribution of shared and special genes of main melanoma states across different publications ([5,7,25,43]): (**A**) the melanocytic (proliferative/differentiated) signatures (Hoek proliferative (Pro) [7], Verfaillie proliferative (Pro) [5], Tsoi melanocytic [25]), and a list of common genes among these signatures. (**B**) The mesenchymal-like (invasive/undifferentiated) signatures (Hoek invasive (Inv) [7], Verfaillie invasive (Inv) [5], Tsoi undifferentiated [25], Rambow invasive (Inv) [43]), and the list of common genes among these signatures.

**Figure 4 cells-11-01157-f004:**
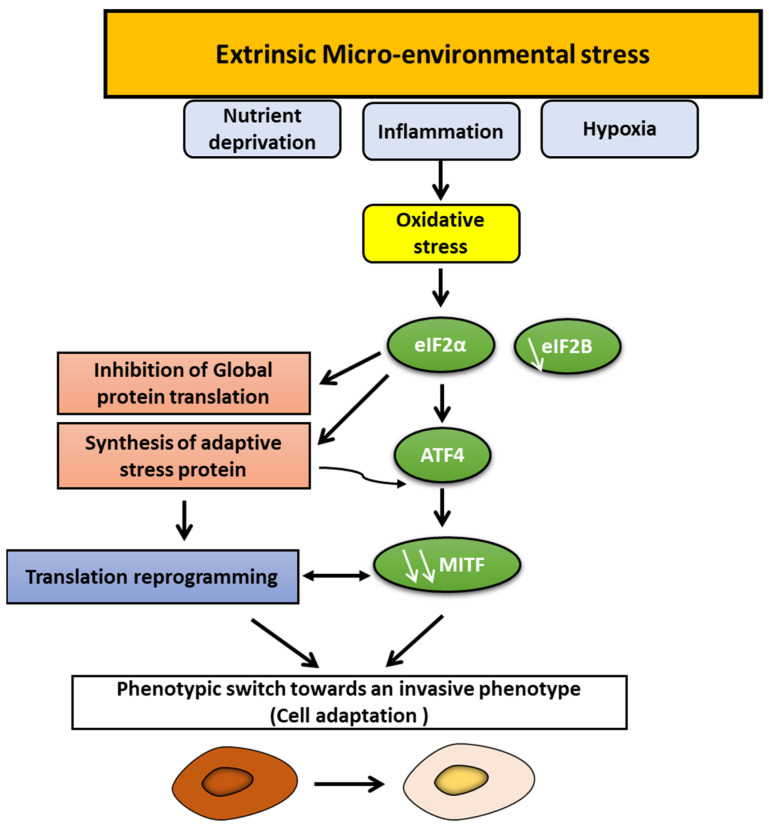
**Phenotype switching in melanoma: prominent role of microenvironment stress signals and oxidative stress.** Several microenvironmental cues, such as hypoxia, inflammation, and nutrient deprivation, can promote oxidative stress leading to eIF2α phosphorylation and eIF2B inhibition and, consequently, to the synthesis of adaptive stress proteins, including ATF4, and to the inhibition of global protein translation, which repress MITF. Translation reprogramming and MITF repression can induce a phenotypic switch towards an invasive state.

**Figure 5 cells-11-01157-f005:**
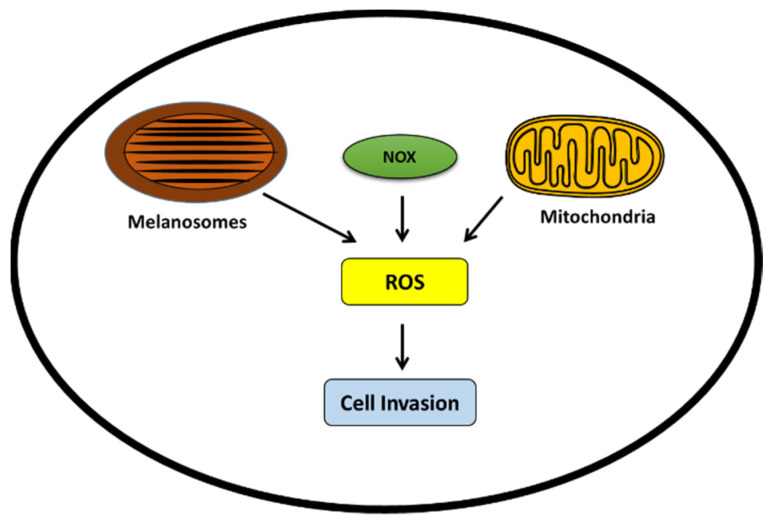
**ROS generation in melanoma cells promoting invasive phenotype.** ROS can be produced by melanosomes, the NOX family, and mitochondria. High ROS levels can promote the invasive phenotype.

**Figure 6 cells-11-01157-f006:**
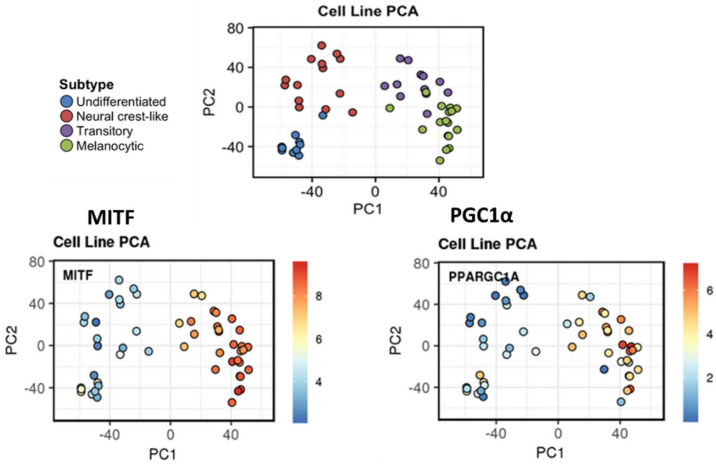
**MITF-PGC1α axis in melanoma plasticity.** High correlation between MITF and PGC1α expression across melanoma phenotypes in the panel of 53 human melanoma lines used in Tsoi et al. 2018 [25]. PCA of melanoma cell line expression profiles are annotated by identified clusters. MITF and PGC1α are highly expressed in the melanocytic (C4-green) and the transitory (C3-purple) phenotypes whereas this expression is low in the invasive phenotypes (neural crest-like (red) and undifferentiated (blue) phenotypes). Illustration was created by using the interactive web interface resource [25] available at http://systems.crump.ucla.edu/dediff/.

**Figure 7 cells-11-01157-f007:**
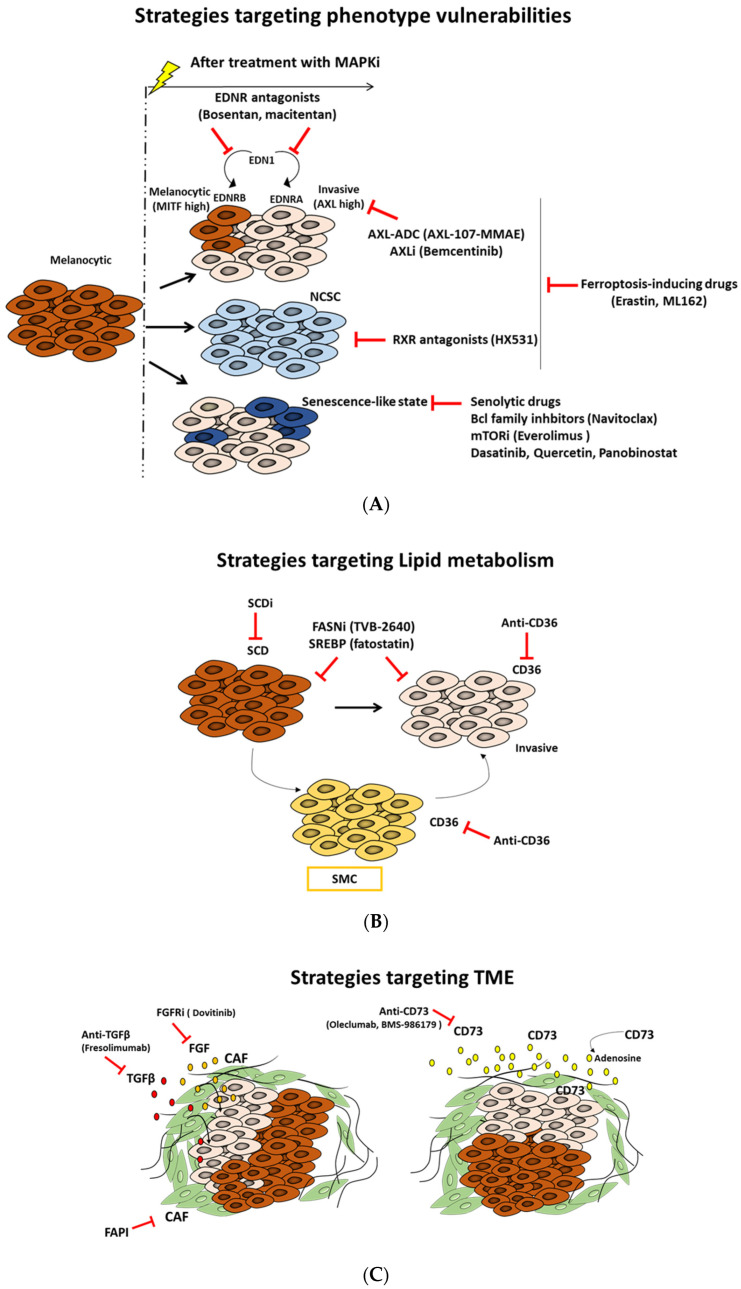
Targeted therapeutic approaches based on melanoma states vulnerabilities, metabolic rewiring, and microenvironment modulation. Schematic description of different therapeutic strategies: (**A**) exploiting vulnerabilities of distinct phenotypes, (**B**) targeting lipid metabolism, and (**C**) modulating components and signals of tumor microenvironment (TME).

**Table 1 cells-11-01157-t001:** Targeting phenotypic plasticity: an overview of innovative and rational approaches for melanoma treatment.

Therapeutic Approach	Targets/Drugs	Stage of Development	References
**Targeting phenotype-specific vulnerabilities**	Invasive phenotype: AXL-(ADC): AXL-107-MMAE Enapotamab vedotin	Solid tumor phases I, II NCT02988817	[23,123]
	Invasive phenotype: Small molecule AXLi (bemcentinib, dubermatinib)	Solid tumor phases I, II Bemcentinib NCT03184571; NCT03649321 Dubermatinib: NCT02729298	[124,125]
	Invasive phenotype: Multi-targeted TKI (sitravatinib)	Solid tumor phases I, II, III Sitravatinib NCT04518046; NCT03666143 NCT04123704;NCT03906071	[126]
	Neural crest stem cell (NCSC) state: pan-RXR antagonist (HX531)	Preclinical	[43]
	Invasive/NCSC phenotypes: ferroptosis inducing drugs (erastin, ML162, and ML21s)	Preclinical	[25]
	Phenotypic heterogeneity/TME: EDNR antagonists	Bosentan NCT04158635	[103]
	Senescence-like phenotype: Senolytics AKT/mTORi HDACi Bcl family inhibitor	Solid tumors phases I, II, III mTORi: everolimus (NCT00876395) HDACi: panobinostat (NCT04897880) Navitoclax (NCT03366103)	[127,128]
**Targeting lipid metabolism**	Lipid uptake: Anti-CD36	Preclinical	[129]
	Lipid synthesis and uptake: SREBPi: fatostatin, botulin, and PF-429242	Preclinical	[130]
	Lipogenesis: FASNi (TVB-2640)	Solid tumor phase II NCT03179904; NCT03808558	[131,132]
**Targeting glutamine metabolism**	Inhibition of GLS1: CB-839	Phase I/II evaluation of CB-839 in combination with nivolumab in melanoma patients (NCT02771626)	[133,134]
**Targeting TME components and signals**	FAPI	Preclinical	[135,136]
	Anti-TGF-beta	Solid tumor phase I NCT00356460	[137]
	Small molecule FGFRi	Dovitinib NCT01831726 NCT01676714	[138]
	Anti-CD73	Solid tumor phases I, II Oleclumab: NCT03611556; NCT03381274; NCT04668300 CPI-006: NCT03454451 BMS-986179: NCT02754141	[139,140,141]

ADC: antibody-drug conjugate; RXR: retinoid X receptor; TME: tumor microenvironment; FASNi: fatty acid synthase inhibitors, SREBPi: sterol regulatory element-binding proteins inhibitors; FAPI: fibroblast activation protein inhibitors.

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
