# Peer review of "Understanding Molecular Mechanisms of Phenotype Switching and Crosstalk with TME to Reveal New Vulnerabilities of Melanoma"

_cells, 2022, doi:10.3390/cells11071157_

Round 1

Reviewer 1 Report

The manuscript entitled “Understanding phenotype switching molecular mechanisms 2 and crosstalk with TME to reveal new vulnerabilities of melanoma” by Najem and colleagues is a well-written and complete review. In this article, the authors bring the main aspects known about melanoma cell plasticity, including the different cell states, their regulatory mechanisms, and current and potential therapeutic strategies targeting the phenotype switch. This is a very relevant area currently investigated, especially in melanoma.

I suggest only minor corrections before the approval of this review, as follows:

  • Page 15, line 589: Th1 means Th1 cells?
  • The legend of figure 6 is incomplete. Please, detail it in order to permit the figure comprehension.
  • In general, the quality of figures could be improved.
  • Page 29, line 20: “…[122]. These…” instead “…[122]. these…”
  • Page 22, line 892: The sentence “…cancer mouse models of [127].” Is truncated. Please, correct.

Author Response

We would like to thank the editor and the reviewers for their time and their efforts. We are so grateful for their positive comments and careful review. The revised manuscript was submitted with tracked changes.

The answers to the reviewer’s comments are as follows:

Author's Reply to the Review Report (Reviewer 1)

  • Page 15, line 589: Th1 means Th1 cells?

Yes, Th1 means “Th1 cells” and the abbreviation was added in the manuscript

  • The legend of figure 6 is incomplete. Please, detail it in order to permit the figure comprehension.

We detailed figure 6 in order to permit the figure comprehension. Modifications are highlighted in the manuscript (Track changes).

  • In general, the quality of figures could be improved

We agree. The quality of figures could be improved, we recognize this issue, and we will download these figures in a separate file in order to preserve the high resolution.

  • Page 29, line 20: “…[122]. These…” instead “…[122]. these…”

Thank you. The error pointed out has been corrected

  • Page 22, line 892: The sentence “…cancer mouse models of [127].” Is truncated. Please, correct.

Thanks for pointing this out. The following sentence was added to replace the truncated one “targeting metastasis-initiating cells by an anti-CD36 antibody showed high efficacy in oral cancer mouse models of human oral carcinomas”.

Reviewer 2 Report

In their paper, Najem et al. discussed the phenotype switching hypothesis in melanoma as well as crosstalk between phenotype switching with TME. In general, the paper is easy-to-read and presents almost all relevant papers. Authors took great efford to summarize the phenotype switching hypothesis and to provide a meta-analysis of the different melanoma signatures. Their paper is of great quality and should be accepted after minor revisions.

1. Lines 42-50 Authors do not discussed the intratumor and intertumor heterogeneity in melanoma, for example genetic heterogeneity, in the introduction. It is crucial for the understanding of phenotype swithing phenomenon in melanoma (f.e. PMID: 29078205).

2. Authors should add the year of the first report of phenotype switching model to the introduction (lines 52-53). Especially since they indicate the year of the Tsoi's subclassification of the melanoma phenotype switching (line 171)

3. Please change MITF-High cells to MITFhigh.

4. Line 202 change "jasper et al, 2020" to "Jasper et al."

5. Figure 1 is interesting, however, it should be divided into 2-3 smaller figures. In this way it would gain clarity.

6. Line 328 "we generated a Venn diagram (....) between these publication". Please cite ALL publications in this sentence that were used to the generation of Venn diagram.

7. "Role of Epigenetic modifications". Authors should mentioned the role of miRNAs in the regulation of phenotype switching, including a putative role of miR-410 in this process (PMID: 32555626).

8. Line 455 "In melanoma" not "In Melanoma"

9. Figure 7 and Table 1 are really great and provide a good overview on the targeting phenotype switching in melanoma.
Minor points:
There are many typos etc. in the manuscript. Moreover, authors often change the way they cite articles (Name et al. year. Name et al year. Name et al; year etc.) Moreover, in some places authors extract new paragraphs of only one or two sentences, which reduces the readability of the manuscript (for example see 5.4.2. CD73). Please correct it in the revised version. 

Author Response

Author's Reply to the Review Report (Reviewer 2)

We would like to thank the editor and the reviewers for their time and their efforts. We are so grateful for their positive comments and careful review. The revised manuscript was submitted with tracked changes.

The answers to the reviewer’s comments are as follows:

  1. Lines 42-50 Authors do not discussed the intratumor and intertumor heterogeneity in melanoma, for example, genetic heterogeneity, in the introduction. It is crucial for the understanding of phenotype switching phenomenon in melanoma (f.e. PMID: 29078205)

We discussed the genetic intertumor and intratumor heterogeneity ((Grzywa et al., 2017). The modifications in the introduction are highlighted in the manuscript (Track changes).

  1. Authors should add the year of the first report of phenotype switching model to the introduction (lines 52-53). Especially since they indicate the year of the Tsoi's subclassification of the melanoma phenotype switching (line 171)

The year of the first report of the phenotype switching model was added to the introduction where appropriate.

  1. Please change MITF-High cells to MITFhigh

MITF-High was replaced by MITFhigh in the whole manuscript

4.Line 202 change "jasper et al, 2020" to "Jasper et al."

Thank you. "jasper et al, 2020" has been replaced by "Jasper et al."

  1. Figure 1 is interesting, however, it should be divided into 2-3 smaller figures. In this way it would gain clarity.

In figure 1, we aimed to present a schematic description of all the aspects of the phenotype switching at the same time (transcriptomic, epigenetic, metabolic, and oxidative stress status). Although, we will download this figure in a separate file to improve the resolution and thereby the clarity.

  1. Line 328 "we generated a Venn diagram (....) between these publications". Please cite ALL publications in this sentence that were used to the generation of the Venn diagram.

As requested, all the publications were cited and citations were added in the sentence.

  1. "Role of Epigenetic modifications". Authors should mention the role of miRNAs in the regulation of phenotype switching, including a putative role of miR-410 in this process (PMID: 32555626).

We add the following paragraph in the revised manuscript (tracked changes). In this paragraph, we explained the role of miRNAs in the regulation of phenotype switching, including the critical role of miR-410 in this process (Grzywa et al., 2020)

“MicroRNAs (miRNAs) are a family of endogenous, small non-coding RNAs that regulate gene expression [65,66]. It has been reported that miRNAs are dysregulated in several cancers including melanoma and they are implicated in several stages of carcinogenesis including both initiation and progression [65,66]. miRNAs can regulate melanoma plasticity. For instance, it was shown that miR-410-3p can up-regulate AXL and drive a phenotypic switch towards an invasive phenotype in melanoma cells [67]. In addition, miR-182 is another key regulator of the EMT that promotes an invasive phenotype by directly repressing FOXO3 (forkhead box O3) and MITF [68]. On the other hand, miR-211 is inversely correlated with invasive phenotype in melanoma [65]. Additionally, miR-542 3p inhibits invasion and EMT in preclinical models of melanoma [65].

Collectively, these results show that miRNAs have a critical role in the regulation of phenotype switching in melanoma.”

  1. Line 455 "In melanoma" not "In Melanoma"

Thank you. The error pointed out has been corrected

  1. Figure 7 and Table 1 are really great and provide a good overview on the targeting phenotype switching in melanoma.

Minor points:

There are many typos etc. in the manuscript. Moreover, authors often change the way they cite articles (Name et al. year. Name et al year. Name et al; year etc.) Moreover, in some places authors extract new paragraphs of only one or two sentences, which reduces the readability of the manuscript (for example see 5.4.2. CD73). Please correct it in the revised version.

Thank you.  These issues (typos, citations, new paragraph of only one or two sentences) were all addressed in the revised manuscript.

Reviewer 3 Report

This is an unique and comprehensive review of the complex nature of melanoma phenotype switching. The article is well written. The results are nicely presented in both text, tables and figures. This article has the potential to become a benchmark reference in future research on melanoma treatment and overcoming therapy resistance. 

Author Response

Author's Reply to the Review Report (Reviewer 3)

We would like to thank the editor and the reviewers for their time and their efforts. We are so grateful for their positive comments and careful review.  

Thank you so much!!!!!. We really appreciate the comments

The revised manuscript was submitted with tracked changes.